# Theoretical Exploring of a Molecular Mechanism for Melanin Inhibitory Activity of Calycosin in Zebrafish

**DOI:** 10.3390/molecules26226998

**Published:** 2021-11-19

**Authors:** Nilupaier Tayier, Ning-Yi Qin, Li-Nan Zhao, Yi Zeng, Yu Wang, Guang Hu, Yuan-Qiang Wang

**Affiliations:** 1School of Pharmacy and Bioengineering, Chongqing University of Technology, Chongqing 400054, China; NILpr@2020.cqut.edu.cn (N.T.); zhaoln@2020.cqut.edu.cn (L.-N.Z.); zy15730447603@hotmail.com (Y.Z.); wangy@2020.cqut.edu.cn (Y.W.); 2Chongqing Pharmaceutical Group Huamosheng Pharmaceutical Science & Technology Co., Ltd., Chongqing 400050, China; vivianqny@hotmail.com; 3Chongqing Key Laboratory of Medicinal Chemistry & Molecular Pharmacology, Chongqing University of Technology, Chongqing 400054, China

**Keywords:** calycosin, tyrosinase, melanin, zebrafish, molecular docking

## Abstract

Tyrosinase is an oxidase that is the rate-limiting enzyme for controlling the production of melanin in the human body. Overproduction of melanin can lead to a variety of skin disorders. Calycosin is an isoflavone from Astragali Radix, which is a traditional Chinese medicine that exhibits several pharmacological activities including skin whitening. In our study, the inhibitory effect of calycosin on melanin production is confirmed in a zebrafish in vivo model by comparing with hydroquinone, kojic acid, and arbutin, known as tyrosinase inhibitors. Moreover, the inhibitory kinetics of calycosin on tyrosinase and their binding mechanisms are determined using molecular docking techniques, molecular dynamic simulations, and free energy analysis. The results indicate that calycosin has an obvious inhibitory effect on zebrafish pigmentation at the concentration of 7.5 μM, 15 μM, and 30 μM. The IC_50_ of calycosin is 30.35 μM, which is lower than hydroquinone (37.35 μM), kojic acid (6.51 × 10^3^ μM), and arbutin (3.67 × 10^4^ μM). Furthermore, all the results of molecular docking, molecular dynamics simulations, and free energy analysis suggest that calycosin can directly bind to the active site of tyrosinase with very good binding affinity. The study indicates that the combination of computer molecular modeling and zebrafish in vivo assay would be feasible in confirming the result of the in vitro test and illustrating the target-binding information.

## 1. Introduction

Melanin pigment has several functions, including protection of the skin from ultraviolet light and removal of reactive oxygen species [1,2]. Nevertheless, excessive production of melanin pigment and its accumulation in the skin can cause pigmentation disorders, such as solar lentigo, melasma, freckles, and post-inflammatory hyperpigmentation [3,4]. Three melanocyte-specific enzymes, tyrosinase, tyrosinase-related protein 1 (TRP-1), and 2 (TRP-2) are involved in melanogenesis, in which tyrosinase is the key enzyme in the melanin synthesis pathway [5,6]. Tyrosinase is a copper-containing enzyme that catalyzes two distinct reactions of melanin synthesis: the hydroxylation of a monophenol and the conversion of an O-diphenol to the corresponding O-quinone. In these oxidation reactions, three different forms of binuclear copper are involved in the active site [7]. In addition, tyrosinase inhibitors have become increasingly important in medicinal and cosmetic products in relation to hyperpigmentation [8]. Therefore, searching for tyrosinase inhibitors with higher bioactivity and lower toxicity has become a hot research topic.

Nature has been a source of medicinal products of many diseases, with a lot of useful drugs developed from plant resources. Polyphenols and flavonoids isolated from natural plants are examples of components that have significant tyrosinase inhibitory effects [9,10]. Flavonoids can effectively inhibit tyrosinase-catalyzed oxidation of L-dihydroxyphenylalanine in cell-free extracts and living cells, and attenuate cell pigmentation induced by the expression of exogenous human tyrosinase [11]. Calycosin (7,3′-dihydroxy-4′-methoxy isoflavone, C_16_H_12_O_5_) (Figure 1) is one of the isoflavones isolated from Astragali Radix, which is a traditional Chinese medicine (TCM) used for immune system enhancement, stamina and endurance increases, and skin whitening for hundreds of years. Calycosin has a wide range of pharmacological activities, including anticancer, anti-inflammatory, anti-osteoporosis, neuroprotection and hepatoprotection, etc. [12,13,14,15,16]. Previous studies have shown that calycosin can decrease melanin production by regulating tyrosinase in vitro [17], however, there is no in vivo study, and the underlining mechanism of action is still unrevealed.

Zebrafish, an effective model system for the high throughput discovery of bioactive small molecules, has primary advantages in the study of disease over rodent models, including hundreds of embryos in a single clutch and optical clarity of the developing embryo, which allows for live imaging at the organism level. In addition, due to the high genetic, physiologic, and pharmacologic similarity with humans [18,19], zebrafish has become an effective model system for the identification of disease-relevant genes and bioactive small molecules [20]. Considering all these advantages, zebrafish is feasible to explore the melanin inhibitory effect of calycosin for human beings based on the following two aspects. Firstly, in our previous research, we found that the dominant phase II conjugation of calycosin in zebrafish larvae matched well with existing knowledge of isoflavone metabolism in mammalians [21]. Secondly, the zebrafish model is already used to evaluate the biosynthesis of melanin regulated by tyrosinase. For example, Chen et al. found that glabridin reversibly inhibited tyrosinase in a noncompetitive manner through a multiphase kinetic process in zebrafish [22]. Yong Tae Jeong et al. reported that sweroside presented inhibition of the body pigmentation and tyrosinase activity in zebrafish in vivo model [23]. For these reasons, zebrafish would be an ideal in vivo model to confirm the potential tyrosinase-inhibiting effect of calycosin.

Molecular docking is one of the drug design methods used to predict the optimal binding mode between a small molecule ligand and a receptor protein. Together with molecular dynamics simulation and free energy landscape analysis, all these molecular modeling and simulation techniques could be applied to obtain details at the molecular level of the inhibitory activities of a certain compound against chosen proteins [24]. Wagle et al. found the specific pharmacophore responsible for the inhibitory activity of calycosin on tyrosinase by docking simulation [25]. However, molecular dynamics (MD) simulations were not performed to calculate the interaction energy between calycosin and tyrosinase.

Above all, calycosin has been proven to exhibit an anti-melanin production effect through inhibiting tyrosinase in vitro. Zebrafish, which shows similarity to humans in many aspects as a whole-organism model, is suitable for evaluating pharmacological activities of small molecules in vivo. Therefore, in our present study, the inhibitory effect of calycosin on tyrosinase is investigated on zebrafish in vivo and is compared to three validated TYR inhibitors that served as the positive control. Besides, molecular docking, molecular dynamics simulation, and free energy analysis are performed to further investigate the binding information between calycosin and tyrosinase.

## 2. Results and Discussion

### 2.1. Inhibitory Effect of Calycosin on Zebrafish 

As shown in Figure 2, the body pigmentation of zebrafish was significantly inhibited in the calycosin, hydroquinone, kojic acid, and arbutin treatment group in a dose-dependent manner. Furthermore, the result of quantitative analysis on the integrated optical density (IOD) value of the whole fish body (Figure 3.) indicated that 7.5, 15, and 30 μM calycosin could significantly inhibit 21%, 26%, and 52% (*p* < 0.001) of the IOD value compared to the blank control group. In addition, the IC_50_ of calycosin, hydroquinone, kojic acid, and arbutin on pigment production inhibition was calculated as 30.35 μM, 37.35 μM, 6.51 × 10^3^ μM, and 3.67 × 10^4^ μM (Table 1), which suggested that calycosin was as effective as Hydroquinone in inhibiting zebrafish pigmentation.

### 2.2. Molecular Dynamics Simulation

To further test the binding pattern between calycosin and tyrosinase, 100 ns molecular dynamics simulations were performed. Root mean square deviation (RMSD) is a measurement of the mean variance of protein structure and original conformation at a given time which is used to monitor whether a complex system reaches stability during simulation. RMSD plots versus time are shown in Figure 4. The RMSD of tyrosinase fluctuated around 1.65 Å after 50 ns, indicating that it reached stability. However, the RMSD of calycosin had a larger shift during MD simulation and stabilized at 1.2 Å in the range of 0–30 ns; then it showed an obvious fluctuation from 30–63 ns. At last, the RMSD of calycosin fluctuated around 0.5 Å after 63 ns, which indicated that calycosin reached a metastable state after undergoing large conformational changes in the simulation process.

### 2.3. Binding Mode and Free Energy Analysis

In order to explore the binding mode between tyrosinase and calycosin, the average structure of the complex from the last 20 ns MD simulation was gained (Figure 5). Figure 5 shows two phenolic hydroxyls of calycosin bound to Met325 (3.3 Å) and Asp336 (2.6 Å) with strong hydrogen bonds, and there was another one between methoxy and Lys79 with a distance of 3.1 Å. The aromatic centchroman of calycosin interacted with Leu327 and Pro338, and the distances were 3.6 Å, 4.1 Å, and 4.5 Å, respectively. Additionally, the benzene ring of calycosin had a weak pi-sulfur interaction with Met325 (5.1 Å).

The MM/GBSA method was used to calculate the interaction energy between tyrosinase and calycosin for the purpose of determining the key residues for tyrosinase complexed with calycosin. As shown in Figure 6, Lys79, Met325, Gly326, Leu327, Asp336, Pro338 had strong interactions with calycosin, mainly due to the van der Waals force, electrostatic interactions, and the contribution of non-polar solvents. Among them, Lys79, Gly326, and Asp336 had low interaction energy, mainly due to the contribution of electrostatic interaction, which was consistent with the hydrogen bond interaction between tyrosinase and calycosin. Secondly, the interaction energy of Leu327 and Pro338 was low, mainly due to the contribution of van der Waals forces, which was consistent with the formation of hydrophobic interaction between tyrosinase and calycosin. In addition, Met325 also had a significant effect on binding energy. 

To predict binding potency for calycosin complexed to tyrosinase, the trajectory of the last 10 ns was selected in the simulation process and the MM/GBSA method was applied to calculate the binding free energy of the complex. Basically, lower binding free energy indicates a more stable system. As shown in Table 2, the calculation results showed that the binding free energy between tyrosinase and calycosin was −52.472 kcal/mol mainly from electrostatic interaction, indicating that calycosin had a good theoretical binding affinity with tyrosinase.

## 3. Materials and Methods

### 3.1. Chemicals and Materials

Calycosin was purchased from Chengdu Gelipu Bio Co., Ltd. (Chengdu, China). Hydroquinone (≥99%, determined by HPLC), arbutin (≥98%), and kojic acid (≥99%) were purchased from a commercial supplier Macklin (Shanghai, China). Ethyl 3-aminobenzoate methanesulfonate was purchased from Aladdin (Shanghai, China). Other regular reagents for zebrafish system maintains were purchased from Wuhan TianZhengYuan Biological Technology Co., Ltd. (Wuhan, China). All buffers and other reagents were of the highest purity commercially available.

### 3.2. Sample Preparation

The stock solution of calycosin (60 mM) was prepared in dimethyl sulphoxide (DMSO), while hydroquinone (2 mM), kojic acid (50 mM), and arbutin (20 mM), which are known as tyrosinase inhibitors, were used in our study as positive controls and prepared in ultrapure water.

### 3.3. Origin and Maintenance of Parental Fish

Wild-type AB strain adult zebrafish (Danio rerio, 4 to 6 months old) were purchased from the Shanghai FishBio Co., Ltd. (China) and maintained in an automated fish housing system at 28.5 ± 0.5 °C under a 14:10 h light to dark cycle and fed freshly hatched brine shrimps three times daily. Embryos were obtained from spawning adults in a breeding chamber overnight with a sex ratio of 2:1 (male to female) according to the standard zebrafish breeding protocol. The embryos were collected within 40 min after the light was switched on and rinsed in E3 medium (5 mM NaCl, 0.17 mM KCl, 0.33 mM CaCl_2_, and 0.33 mM MgSO_4_, and pH 7.2) at 28.5 ± 0.5 °C.

### 3.4. Zebrafish In Vivo Assay

The collected synchronized zebrafish embryos were arrayed by pipette into a 12-well plate, 25 embryos per well with 3 mL embryo medium. The prepared solutions of calycosin, hydroquinone, kojic acid, and arbutin were added to the embryo medium from 24 to 48 hpf (hours post fertilization, total 24 h exposure), and the medium concentration of each treatment group was: calycosin, 7.5 μM, 15 μM and 30 μM; hydroquinone, 40 μM, 80 μM and 160 μM; kojic acid, 1.0 μM, 5.0 μM and 50 μM; arbutin, 1.0 μM, 10 μM and 20 μM. Phenotype-based body pigmentation assessment was performed at 48 hpf. The embryo was anesthetized in ethyl 3-aminobenzoate methanesulfonate, fixed on a recessed glass slide with gelatin, and observed under a stereomicroscope Z16 (Leica Microsystems, Ernst-Leitz-Strasse, Germany). The intensity of pigmentation in zebrafish was evaluated according to the integrated optical density (IOD) value of the whole fish measured by Image-Pro plus 6.0.

### 3.5. Molecular Docking Studies

Molecular docking was performed to investigate interactions between calycosin and tyrosinase. Firstly, the crystal structure of tyrosinase was downloaded from the protein data bank (PDB ID: 2Y9X) (http://www.rcsb.org/, accessed on 8 November 2021) [26], and all of the water and metal ions were removed. Secondly, the binding site of tyrosinase was defined based on co-crystal ligand after adding hydrogens and charges. Thirdly, a standard molecular docking was performed by Surflex-Dock in Sybyl 2.0 with the default parameter setting. The best-docked pose with the lowest RMSD to co-crystal ligand was selected to construct complex with tyrosinase for subsequently MD simulation. The binding visualization was performed utilizing PyMOL software.

### 3.6. Molecular Dynamic (MD) Simulations

The MD simulations were carried out using the AMBER16 package. Before the simulation, the molecular mechanics method was adopted to optimize the biological macromolecular system. The general AMBER force field (GAFF) was used for the inhibitors, and the ff99SB force field was employed for the protein [27]. All protein inhibitor complex systems were immersed in a box of the TIP3P water model [28]. The systems were then neutralized by the addition of Na^+^ or Cl^−^ counter ions. Firstly, several minimization steps were performed for the systems to avoid possible steric crashes. Then, each system was gradually heated from 0 K to 300 K during the heating stage and kept at 300 K during the following equilibrium and production stages. A time step of 2 fs was used for the heating stage, equilibrium stage, and the entire production stage. A periodic boundary condition was employed to maintain constant temperature and pressure (NPT) ensembles. The pressure was set at 1 atm and was controlled by an anisotropic (x-, y-, z-) pressure scaling protocol with a pressure relaxation time of 1 ps. The temperature was regulated using Langevin dynamics with a collision frequency of 2 ps-1. The Particle Mesh Ewald (PME) method was adopted to handle long-range electrostatics and a 1 nm cutoff was set to treat real-space interactions. All covalent bonds involving hydrogen atoms were constrained using the SHAKE algorithm. Each system underwent 50 ns MD simulation, and the trajectory of simulated systems was saved every 100 ps.

### 3.7. Free Energy Analysis

For the saved MD simulation trajectories, the MM/GBSA and MM/PBSA methods were used to calculate the binding energy of receptors treated with calycosin [29]. A total of 200 snapshots were taken from 80 to 100 ns to calculate the average binding energy as follows:ΔE_bind_ = ΔE_MM_ + ΔE_SOL_ = ΔE_MM_ + ΔE_GB_ + ΔE_SA_
ΔG_bind_ = G_complex_ − G_protein_ − G_ligand_
=ΔH − TΔS ≈ ΔG_gas_ + ΔG_sol_ − TΔS
ΔG_gas_ = ΔE_ele_ + ΔG_vdw_; ΔG_sol_ = Δ_GPB/GB_ + ΔG_SA_

In the formula above, ΔG_bind_ was the final binding free energy. G_complex_, G_protein_, and G_ligand_ were the free energy of the complex, tyrosinase protein, and ligand, respectively. ΔG_gas_ was the gas-phase interaction energy between a protein and a ligand, consisting of ΔE_ele_ (electrostatic energy) and ΔE_vdw_ (van der Waals force). ΔG_sol_ was the solvation free energy, which was composed of the electrostatic solvation energy ΔG_PB/GB_ (polar contribution) and the non-static solvation energy ΔG_SA_ (non-polar contribution). In addition, the energy of each residue was broken down into the main chain and side chain atoms, and the energy decomposition could be analyzed to determine the contribution of key residues to binding [30].

### 3.8. Statistical Analysis

All data were presented as mean ± standard deviations (SD) of at least three different experiments. Multiple group comparison was conducted by one-way analysis of variance (ANOVA) of IBM SPSS Statistics 19 (version 24, SPSS, Inc., Chicago, IL, USA). A *p*-value of less than 0.05 was considered statistically significant.

## 4. Conclusions

In this study, a rapid and effective method for tyrosinase inhibitors screening was established based on zebrafish in vivo assay and computational studies including molecular docking, molecular dynamics simulations, and binding free energy calculations. Calycosin, an isoflavone from Astragali Radix, exhibited competitive inhibitory activity on melanin production in zebrafish compared with hydroquinone, kojic acid, and arbutin, which are known as tyrosinase inhibitors. In addition, computational studies showed that three parameters including van der Waals force, electrostatic interaction, and non-polar solvents of the key residues MET325, GLY326, LEU327, ASP336, and PRO338 contribute to the good binding affinity between calycosin and tyrosinase. Our study demonstrated a feasible approach for the screening of the pharmacological effect of small molecules by combining the zebrafish in vivo model with molecular docking techniques. 

## Figures and Tables

**Figure 1 molecules-26-06998-f001:**
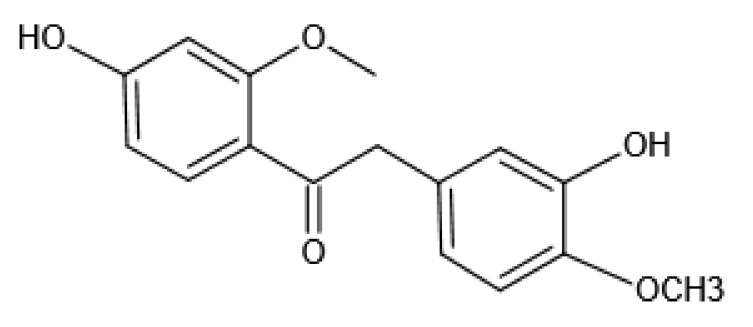
Structure of calycosin.

**Figure 2 molecules-26-06998-f002:**
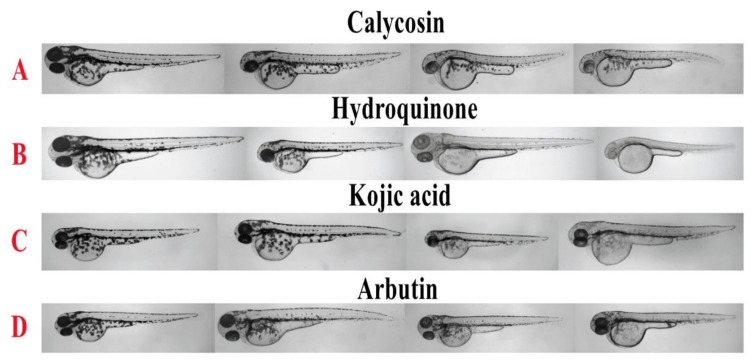
Results of zebrafish pigmentation inhibition. The effects of calycosin and tyrosinase inhibitors (hydroquinone, kojic acid, and arbutin) on the pigmentation of zebrafish were observed under a stereomicroscope. From left to right, the concentrations were 0, 7.5, 15, and 30 μM for calycosin (**A**); 0, 40, 80, and 160 μM for hydroquinone (**B**); 0, 1.0, 20, and 50 mM for kojic acid (**C**); and 0, 1.0, 10, and 50 mM for arbutin (**D**).

**Figure 3 molecules-26-06998-f003:**
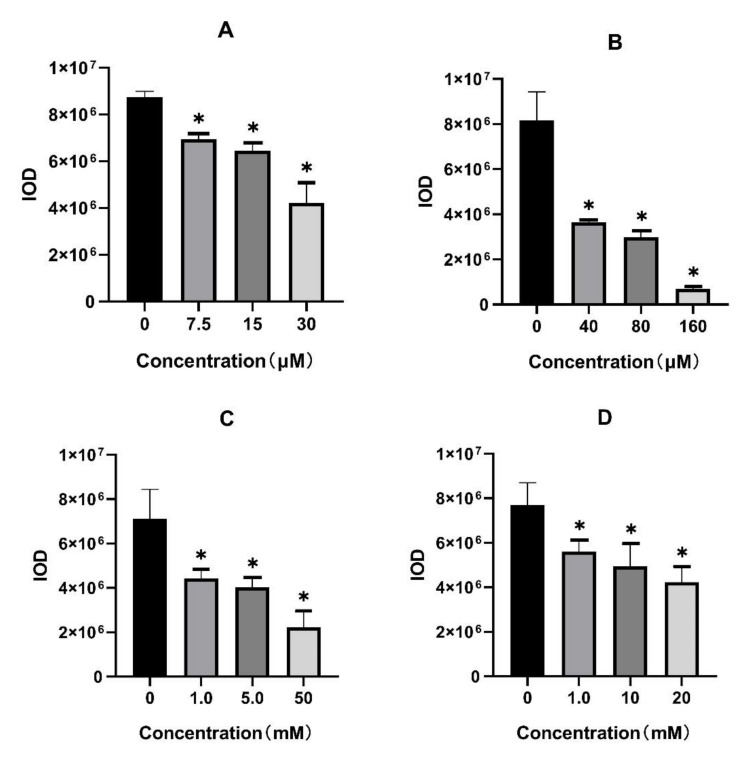
The IOD value of zebrafish in calycosin (**A**) and tyrosinase inhibitors (hydroquinone (**B**), kojic acid (**C**), and arbutin (**D**)) treatment group. * *p* < 0.05 versus blank control group.

**Figure 4 molecules-26-06998-f004:**
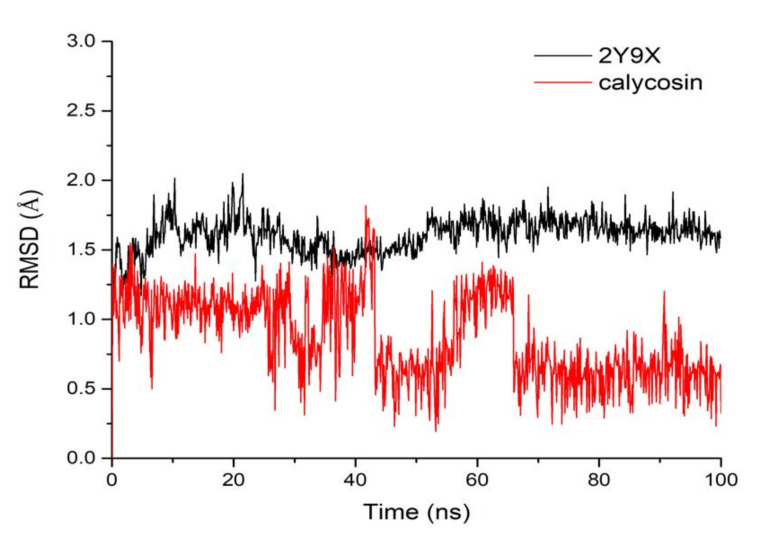
RMSD of tyrosinase (2Y9X) and calycosin complex.

**Figure 5 molecules-26-06998-f005:**
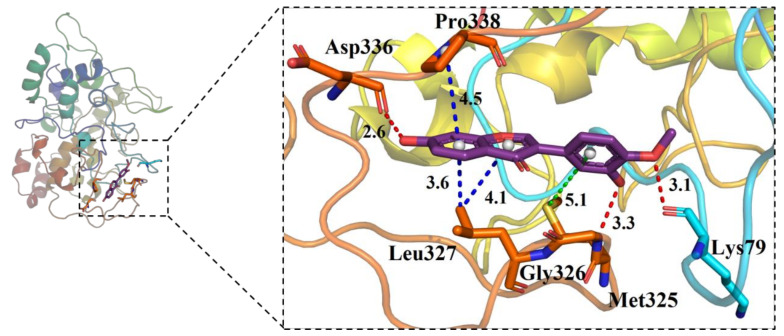
Interaction diagram of tyrosinase (2Y9X) and calycosin after 100 ns molecular dynamics simulation. Red represents the hydrogen bond, blue represents the hydrophobic bond, and green represents the Pi-Sulfur bond.

**Figure 6 molecules-26-06998-f006:**
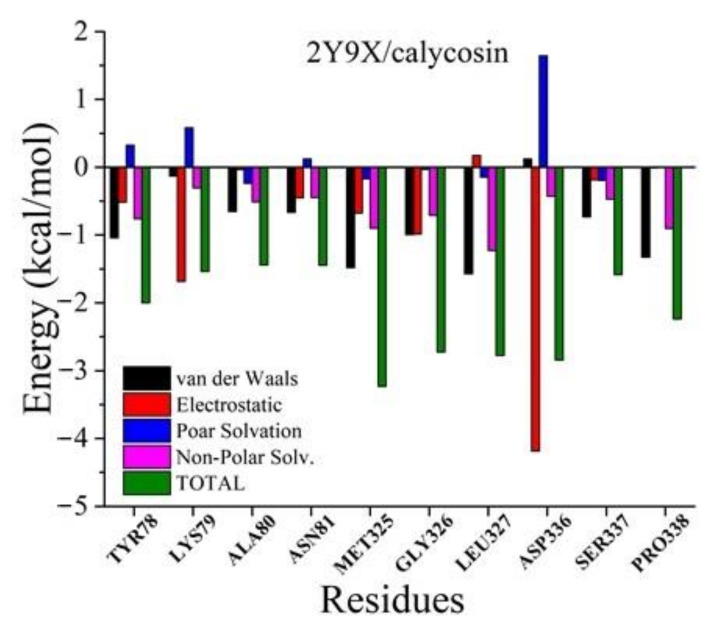
Free energy contribution of 2Y9X critical residues in the binding between 2Y9X and calycosin obtained with MM/GBSA method.

**Table 1 molecules-26-06998-t001:** The IC_50_ value of calycosin and tyrosinase inhibitors.

Compound	IC_50_ (μM) of Pigmentation Inhibition
Calycosin	30.34
Hydroquinone	37.35
Kojic acid	6510
Arbutin	36720

**Table 2 molecules-26-06998-t002:** Binding free energy and single energy (kcal/mol).

Method	ΔG_vdw_	ΔG_ele_	ΔG_pol_	ΔG_non-pol_	ΔG_bind_
MM/GBSA	10.553	−62.980	−9.905	10.004	−52.427

## Data Availability

The data presented in this study are contained within the article.

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
