# Peer review of "Theoretical Exploring of a Molecular Mechanism for Melanin Inhibitory Activity of Calycosin in Zebrafish"

_molecules, 2021, doi:10.3390/molecules26226998_

Round 1
Reviewer 1 Report
I do not see what is the relationship between the experimental zebrafish assay and the simulation study using molecular dynamics. The experimental part only indicates that there is a blocking effect, while any parameter which was measured experimentally was not compared to a parameter estimated by MD. Such type of a research work can be done, for example, using NMR spectroscopy. In this respect, I do not understand what is the aim of the study since the selection of parameters in the MD simulation is critical for the trueness of the final result and without an experimental support, the latter can be the result of speculation.
The results of the statistical analysis in the zebrafish assay are also not clear. The ANOVA analysis should be done for all of the levels of the factor and then a multiple comparison by Bonferroni or other correction should be performed. One level of significance should be used. ‘The lower P-value, the better’ is a wrong practice in presenting the results of any study and shows misunderstanding in the use of P-values.
Reviewer 2 Report
Article molecules-1428342-peer-review-v1
Theoretical Exploring of Molecular Mechanism for Melanin In-hibitory Activity of Calycosin in Zebrafish
The paper makes a significant and interesting contribution to future readers of the journal.
Some minor suggestions are given below, after which the paper should be considered for acceptance.
- Please insert figure of calycosin according Jin Hee Kim,Mee Ree Kim,Eun Sook Lee,Choong Hwan Lee. Inhibitory Effects of Calycosin Isolated from the Root of Astragalus membranaceus on Melanin Biosynthesis [J]. Biological and Pharmaceutical Bulletin 2009, 32(2).
- Please check the format of the bibliographic references, some of them do not agree with the journal, for example Jin Hee Kim et al2009
- Author 1, A.B.; Author 2, C.D. Title of the article. Abbreviated Journal Name Year, Volume, page range.
Reviewer 3 Report
1. How to choose the dose of the drugs?
2. The effect of Hydroquinone is better than Calycosin.
3. What is IOD? Please clearly describe the IOD.
Round 2
Reviewer 3 Report
The revised manuscript can be accepted.